# Benchmark Approach for Efficiency Improvement in Green Supply Chain Management with DEA Models

Farzad Zaare Tajabadi and Sahand Daneshvar *

Department of Industrial Engineering, Engineering Faculty, Eastern Mediterranean University, Famagusta 99628, North Cyprus Via Mersin 10, Turkey
* Correspondence: sahand.daneshvar@emu.edu.tr

**Abstract:** Nowadays, concerns about environmental issues are increasing. Therefore, companies and producers are under pressure from government rules and regulations on one hand, and on the other hand, maintaining customer satisfaction concerning cares about the environment. Green supply chain management (GSCM) is a procedure to increase efficiency and decrease environmental effects for companies that collaborate with customers and suppliers. According to GSCM, there is some research about applying green aspects of purchasing, design, manufacture, distribution, packaging, marketing, and reverse logistics of supply chains to improve their company's performance regarding environmental issues. Moreover, recently, DEA as a nonparametric model is used to evaluate the efficiency and performance of supply chains as decision-making units (DMUs). However, previous studies on efficiency improvement in GSCM did not investigate the effect of some economic and environmental factors together such as service level, emissions ($CO_2$), and size of the supply chains (arcs) on the efficiency of the whole supply system. These factors are essential as they can affect the manager's ability to distinguish the true performance of a green supply chain. Thus, evaluating the efficiency of GSCM by DEA models and imposing the green principles to find out the efficient ones for increasing management performance is vital. Fulfilling the mentioned research gap, this paper developed a benchmark approach to verifying efficient DMUs and potential efficient DMUs which may improve costs and efforts to become efficient. In the case study, the benchmarks and potentially efficient DMUs are found by DEA standard models and slight adjustment is conducted for potentially efficient DMUs to change their status to efficient DMUs. Moreover, the effect of some green principles on the efficiency value of DMUs is verified using Tobit regression before and after the mentioned modification. A set of realistic results provided for the priority of potential DMUs modification confirmed the applicability of the proposed procedure.

**Keywords:** green supply chain management; performance evaluation; efficiency; benchmarking; data envelopment analysis; Tobit regression

## 1. Introduction

With the increasing concerns about the environment in the recent decade, environmental pollution should be paid attention to in processes besides industry development. The environmental and economic benefits of product remanufacturing, as well as consumer environmental consciousness, have pushed numerous manufacturing and retail enterprises to produce and sell green products [1]. All the solutions to this problem should be combined and reviewed in a comprehensive supply chain procedure framework. Supply chain management (SCM) is an important factor that is directly related to the company's efficiency and competitive position. SCM entails planning product manufacture from raw materials to customer delivery [2]. Supply chain management is an essential component of every company organization, and proper planning will assure economic, environmental, and social sustainability [3]. Greening the supply chain is a new concept. Green supply chain management (GSCM), in particular, has seen considerable expansion in recent years [4].

Green supply chains rely primarily on activities that cut costs, enhance efficiency, and generate "green" or environmentally friendly products [5]. According to this concept, the purchaser uses his/her purchasing power to demand better environmental performance from the upstream supplier in the supply chain. This means that in most cases, the purchaser is a big company that has a facilitating role for its suppliers. These suppliers are usually companies of small or medium sizes and they help them to become environment-friendly organizations. In recent years, following the rapid industrialization of most of the developed economies, environmental losses have been paid a great deal of attention and all over the world governments have begun to apply environmental protection rules. Although from a merely economic view and regardless of some side factors, industry and industrial activities, constituting a major share of national income and a large percentage of the present human force of the sociality, are considered the main basis of development and growth of the countries and also as a major indicator of development. From a sustainability view, the benefits of such a development, which results from not regarding environmental aspects of exploitation of national resources and environmental protection, are not justifiable in the long term; from an environmental view, destructive consequences of industrial activities will finally lead to health damage, lack of working motivation, and above all, gradual or sudden decrease in life quality in societies and residential. This imposes further costs on society in addition to the apparent and tangible costs.

In latest years, nearly 70% of the world's leading firms have prioritized sustainability in their work plans [6]. According to the reports of leading firms, the success of their sustainability activities is also dependent on collaboration with supply chain (SC) participants [7]. Green supply chain management (GSCM) entails incorporating environmental and economic goals into the supply chain's operating plan. This type of integration reduces carbon footprint while enhancing financial return and profitability [8]. Bowen et al. [9] define GSCM as the "integration of the company's purchase plan with the environmental activities in SCM, to improve the environmental performance of supplier and customers." Concerns of product design, usage, reuse, disassembly, and final disposal are also included in GSCM [10], in addition to storage, transportation, supplier development to satisfy green purchasing criteria, and encouragement for the adoption of environmental certifications such as ISO 14000 [10,11]. Zhu and Sarkis [12] consider GSCM to be the integration of environmental thinking with SC operations management, beginning with product design and progressing through raw material selection, manufacturing processes, transportation and delivery, and the end consumer arriving at the final destination after usage. According to the definition of Large and Thomsen [13], GSCM comprises the design process, raw material selection, green procurement, green production, green distribution, and reverse.

Therefore, regarding the emphasis on organizational efficiency to effectively and properly use the resources for achieving organizational goals, and also due to the requirements of national and international rules about environmental issues, proper compromise and compatibility between the two goals of economic growth and environmental protection, and integrating the two important issues of efficiency and environmental protection under the title "green efficiency program" can be of great importance.

This study poses the following research question: how we can measure efficiency in green supply chains? How we can distinguish the potential DMUs among inefficient ones that can be efficient with less cost and effort according to benchmark DMUs? How can we verify the effect of green factors on the decision-making unit's efficiency values?

The following can consider as the contributions of the current study. Efficiency evaluation of the SCs as DMUs with a different product, production cost per unit, and chain size to introduce the efficient DMUs and consider them as a benchmark. Finding the potential DMUs among the inefficient DMUs who have the ability to become efficient after modification according to the benchmarks. Finally, verify the effectiveness of three green factors such as service level, emissions ($CO_2$), and the size of the chains (arcs) on the efficiency of the whole system before and after modifications of the potential DMUs and compared the results. The expectation is the positive effect of service level (customer

satisfaction) and arcs (as the size of the transportation factor) and the negative effect of $CO_2$ emission (environmental effect) on the efficiency value of DMUs. Moreover, the changes in the mentioned effects after improvement adjustments will help the decision-makers to identify the true performance of the green supply chains.

The rest of the article is organized as follows: The next part briefly explains the relevant research. Section 3 is introduced the methodology procedure. In Section 4 a case study is considered and some analytical discussions are provided. The conclusion and result are performed in Section 5 with guidelines for future studies.

## 2. Literature Review

In this section, backgrounds on supply chain management, methods of solving multi-objective problems of supply chain network designing, green supply chain management, developed benchmark approach, and standard DEA models are discussed for evaluating the efficiency of supply chains.

### 2.1. Background of Supply Chain Management

In the 1990s, along with the improvement of production capacities, the managers of industries found that the raw materials received from different suppliers have a significant role in increasing organizational capabilities to meet the customers' needs. This also significantly affected the organization's concentration, supply bases, and resourcing strategies. Additionally, the managers found that only manufacturing a qualitative product is not enough, and in fact, supplying the products with the criterion considered by the customers (when, where, and how) and their desired quality and cost led to new challenges. In such conditions, as a conclusion of the mentioned changes, they found that they will not be enough to manage their organization for a long time. In managing the network, they shall be involved with the customers in managing the network of all the factories and companies that supplied their organization inputs directly or indirectly and also the network of companies engaged in delivery and after-sale services. With such an attitude, approaches of "supply chain" and "supply chain management" appeared.

Supply chain management (SCM) is considered one of the important management activities in most organizations. In general, all of the activities of an organization that associate the suppliers, factories, warehouses, retailers, and final consumers and require managing the goods, finances, and information are referred to as supply chain (SC) in the literature [14]. Due to the complexity of mutual relationships between different components of SC, the supply chain is exposed to a wide range of risks and this makes decision-making a challenging problem for the managers. Undoubtedly, one of the most important and difficult aspects of the supply chain structure of goods and services is making decisions about facility location for each layer of the chain. This is considered one of the strategic decisions; usually, its change is not justifiable in the short term or even midterm period [15]. There are examples that there is uncertainty and risk in the reliability of chain facilities due to phenomena such as natural disasters, changes in capital holders, mistakes of the labour force, climatic conditions, etc. In October 2001, the prevalence of anthrax virus in the Washington branch of the US postal service led to the closure of the 633,000 square feet office (facility) and as a result, the loss of a major part of the capacity of the chain. Hurricane Katrina in 2005 resulted in the fact that after many years, parts of central Louisiana are still faced with a shortage of materials due to the breakdown of some factories and warehouses [16]. Quarrels of labour forces in September 2002 led to the closure of western ports of America, whereas the retailers intended to store a great deal of depot before the beginning of seasonal holidays. In this situation, production was stopped for a while in some factories. Therefore, when making strategic decisions, uncertainties of the real world should be possibly considered, so that at the time of their occurrence, the system can continue its functioning with the minimum loss.

### 2.2. Background of Research on Methods of Solving Multi-Objective Problems of Supply Chain Network Designing

There are few types of research in the literature on supply chain design with a multi-objective optimization approach. Sabri and Beamon [17] used the ε limitation method for solving the model after designing an integrated three-purpose model for minimizing the cost, maximizing the service rate, and flexibility about the uncertainty of delivery time and demands for the product. Chan et al. [18] proposed a multi-objective genetic algorithm for demand-oriented distribution problems in the supply chain network. The goals of the mentioned problem included optimization of system costs, total delivery time, and an efficiency rate of the manufacturers' capacity. Chen and Lee [19] designed a multi-period, multistage, and multiproduct scheduling model in a supply chain network about the uncertainty of demand and product price with the goal of fair distribution of profits among all the stakeholders, keeping the inventory and service rate at the optimum level, and stable decision making regarding the instability of demand. For solving the problem, they used the two-stage fuzzy decision-making method. By using limitation ε and branch and bound techniques, Guillén et al. [20] solved their two-purpose model (to maximize the profit in the determined period and increase the level of customer satisfaction) which belonged to the category of linear mixed-integer random planning.

Considering operational costs, service level, and resource efficiency as the goals of this study of production and distribution problems, Chan et.al. [21] proposed a hybrid approach based on the combination of genetic algorithms by the hierarchical method (AHP).

### 2.3. Background of Green Supply Chain Management

Green supply chain management was introduced by the industrial research association of Michigan State University in 1996. It is a new management model for protecting the environment. From the perspective of the product life cycle, green supply chain management includes all the stages of raw materials, designing, manufacturing the product, selling the product, transportation, using the product, and recycling the product. Using green supply chain management and technology, the company can decrease the negative environmental effects and achieve the optimum use of resources and energy. Greening the supply chain is the process of considering the environmental criteria or observations throughout the supply chain. Green supply chain management integrates supply chain management with environmental requirements in all the stages of designing the product, choosing, and greening the supply chain, considering environmental criteria or observations throughout the supply chain. Green supply chain management integrates supply chain management with environmental requirements in all the stages of designing the product, choosing, and supplying the raw materials, manufacturing, and building, distribution and transportation processes, delivering to the customer, and after consumption, managing recycling for maximizing the efficiency of energy and resource consumption besides improving the function the whole supply chain [22]. In reviewing the environmental effects of supply chain activities, the effects of the products on the environment are analysed by a holistic approach (including the analysis of the product life cycle from the beginning to the end of its life). In this approach, all the ecological effects (the science of the creatures' habits and life and their interaction with the environment) of every activity in different stages of the product life such as the concept of the product, designing, preparing raw materials, manufacturing and building, montage, keeping, packing, transporting, and further use of the product are measured and considered in designing the product [23].

Kuo et al. [24] want to create a green supplier selection model that combines an artificial neural network (ANN) and two multi-attribute decision analysis (MADA) methods: data envelopment analysis (DEA) and analytic network process (ANP). Hsu and Hu [25] introduced 19 environmental criteria in their article and classified them into five groups. They considered five groups purchase management, research and development management, process management, quality control of the input materials, and system management, and then, they chose the suppliers using the network analysis process technique.

In their study, Chen et al. [26] chose 18 criteria, the most important of which include environmental criteria, environment management system, supplier's profitability, and close relationships of the supplier, and then, using fuzzy theory, the criteria were converted to definite numbers.

In a case study on the printed circuit board in Taiwan, Chen et al. [26] sought for implementing green supply chain management for selecting the supply. He developed two classes of environmental and non-environmental criteria in the studied company, determined the weights of the criteria based on qualitative and quantitative factors, and finally used the Gray analysis method for rating the suppliers. Shaik and Abdul-Kader [27] used a framework constituted of environmental criteria, green criteria, and organizational criteria for selecting the green supplier. He created a hierarchy for evaluating the criteria and sub-criteria of suppliers which led to the compilation of an appropriate liable strategy by the managers. For a two-echelon supply chain, Tajabadi and Kazemi [28] presented an NL-IP model aiming to minimize total costs, maximize demand served, and minimize transportation pollution. Two meta-heuristic algorithms, NSGA-II and NRGA, are developed to solve the problem, and the Taguchi method is used to set the parameters. Forghani et al. [29] modified classic stochastic data envelopment analysis (SDEA) model by manipulating weak efficient hyperplanes. The suggested model was applied on environmental efficiency of sustainable development goals in Latin America and Caribbean (LAC) countries resulting in better discrimination. A summary of the research on green supply chain management is presented in Table 1.

**Table 1.** Criteria of the green supply chain in the previous studies.

| Criteria | Component | References |
|---|---|---|
| Green supply and purchase | Choosing the supplier regarding the environmental criteria, providing the materials in environment-friendly packs, having environmental certificates such as ISO 14000, holding seminars for informing the suppliers about environmental issues, supporting the suppliers in improving their environmental performance, and requiring the suppliers to observe environmental rules, and buying recyclable materials | [24,30–35] |
| Green designing | Designing the products regarding the reduction in material or energy consumption, recyclability of the products, designing the product for reducing or avoiding the consumption of dangerous materials or inappropriate production processes, designing the products to reduce their environmental effects | [24,30,31,36–39] |
| Green production | A commitment of the senior and junior managers to observe the environment-related rule have qualified environmental management, have environmental certificates such as EUP, ROHS, and ODC, use materials with less harm to the environment, use devices with less pollution to the environment, control release of dangerous gases such as ammonia and $CO_2$, using appropriate methods for removing wastewater, having an appropriate environmental position to other manufacturers, low occurrence of environmental incidents, decreasing noise pollution, holding environmental educational programs for the staff and the managers, focusing on reducing the wastes and optimizing the use of materials, using environment-friendly equipment and technology | [24,25,31,34,36–40] |
| Green packing | Using recyclable packs and containers, using environment-friendly materials in packing the products, using labels for showing the level of accordance of the product with environmental standards, using labels for showing the recyclability of the product | [24,30,31,34,36–39] |
| Green transportation and distribution | Marketing the products relies on environmental issues such as emphasizing environmental certificates, increasing the consumers' environmental awareness, choosing clean transportation methods, returning the products to the company for recycling, better competitive situation than other competitors, choosing the distribution networks and customers with an emphasis on environmental criteria | [24,30,31,34,36–39] |
| Green production costs | The costs of eliminating dangerous and harmful materials, costs of producing environment-friendly products, costs of offering environment-friendly packs, costs of informing the staff about environmental products | [24,30,31,36,38,39] |

According to Table 1, this study emphasis on three criteria in green supply chain management such as green transportation, green production, and green production cost in order to reduce the emissions and cost of production by considering some green factors' efficacy on the whole system's efficiency like service level to meet the customer satisfaction about

the ordering green product cost and delivery, emission ($CO_2$) caused by transportation vehicles through chain and arcs which indicates the size of the chain for transportation.

### 2.4. Background of Benchmarking

Benchmarking is the process of searching for best practices and trying to emulate them [41]. Benchmarking has quickly become a standard practice among top companies. GSCM benchmarking is the practice of comparing a company's green goods, services, and procedures along the supply chain to the relevant indicators of successful enterprises or chains. As a result, GSCM benchmarking encompasses a wide variety of factors like processes, products, performances, and strategies. Data envelopment analysis (DEA) is a widespread and accepted tool for measuring efficiency and performance for many years. According to Stewart [42], one of the standard outputs of DEA can be a benchmark for inefficient DMU which by slight implication can reach the desired point. In the standard DEA models, mainly the inefficient DMUs reveal the previous data. So, it means that if we have selected efficient DMUs as a benchmark, they are suitable for now and maybe not inefficient in the future. Da Costa et al. [43] recognized sustainability indexes for benchmarking the performance and decreasing the environmental effects of the product life cycle with a benchmarking method. Radovanov et al. [44] operated a two-stage DEA model for benchmarking and improving the sustainability performance of tourism-driving services. A majority of sustainability research has examined the relationship between energy, environmental, and economic factors while focusing less on how social factors impact safety performance [45,46].

### 2.5. Background on the Application of DEA in SCM and GSCM

Data envelopment analysis (DEA) is an implementation for measuring the efficiency of decision-making units (DMU) with multiple inputs and outputs. Thus, in the past decade, it has been used for many goals like in the economy, environmental issues, selecting the best supplier, and many more. According to Lambert et al. [47], DMUs impart on several bases like transport, business, universities, hospitals, etc. Classic DEA considers DMUs as a black box which their structures denied, and performance evaluation of DMUs is just related to the inputs and outputs. However, in most cases, DMUs have a network structure, for example that the output of the first stage will be the input for the next stage. Färe and Grosskopf [48], introduced the overall efficiency measurement of new methods with DEA models. Also, overall efficiency can be considered averagely weighted regardless of the state of the efficiency for each stage. Thus, in recent years, many studies were conducted on DMUs that are considered efficient for each stage. DEA is recommended to assist in traditional benchmarking activities and to give management guidance [49]. According to different experiences, this method is an effective way of evaluating performance, benchmarking, and improving the company's performance. In consequence, since DEA is first proposed by Charnes et al. [50], it is widely used in benchmarking studies. DEA also demonstrates a positive impact on defining functions and operating efficiency of different firms [11].

Evaluation of the efficiency of green supply chain management should consider the effect of environmental factors besides the profitability of the firms. These factors may lead to the failure of green supply chains in meeting the green principles and the expectation of customers who care about the environment. Hence, regular evaluation of the efficiency of green supply chains could impact the performance regarding environmental issues. When these factors are considered simultaneously, green supply chain management can be planned more efficiently to minimize the environmental effects and production costs. Thus, the aim of this work is to present a new approach to evaluating the efficiency of green supply chains when some of them are inefficient and has the potential to become efficient through their benchmarks, and the environmental factors are considered. An important novelty in this approach is by benchmarking the efficient DMUs we are able to distinguish the potential DMUs and modify them to become efficient while they are following the

green principles factors at the same time. In addition, the comparison of results before and after modification of potential DMUs through green factors and their effect on the whole system efficiency has been applied.

## 3. Method

### 3.1. DEA Efficiency Analysis

DEA is a method for evaluating efficiency that employs a nonparametric approach and has been widely applied in operations research and general management [50]. Productivity is a function of economic factors that determine output and input [51]. It is important to identify both desirable and undesirable outcomes in environmental evaluation [52]. Based on the analysis of more than 100 energy and environment research papers, Zhou et al. [53] identified five efficiency measurements: (1) radial efficiency, (2) non-radial efficiency, (3) slacks-based efficiency, (4) hyperbolic efficiency, and (5) directional distance function efficiency.

We assume that each DMU $j$ has multiple inputs $x_{i,j}$ and multiple outputs $y_{k,j}$. The relative efficiency measure is defined as follows:

$$\text{Efficiency} = \frac{\sum_k u_k y_{k,j}}{\sum_i v_i x_{i,j}} \tag{1}$$

where $u$ and $v$ are weights. The efficiency is often scaled to range between 0 and 1.

The weights have a flaw: assigning a standard value to them across all DMUs is quite arbitrary. The key notion underlying DEA is that we give each DMU $j_0$ the ability to select its weights. It can do so by solving the following optimization issue. Maximize the efficiency of DMU $j_0$ while keeping all other DMUs' efficiencies less than or equal to 1 [54].

$$\text{Maximize } \theta_0 = \frac{\sum_k u_k y_{k,j0}}{\sum_i v_i x_{i,j0}}$$
$$\text{Subject to } \frac{\sum_k u_k y_{k,j}}{\sum_i v_i x_{i,j}} \leq 1 \ \forall j \tag{2}$$
$$u_k, v_i \geq 0$$

It is not an LP model. So, make a simple change to fix the denominator to a constant value of 1, which can be set as a constraint on the $v_j$ weights as below:

$$\text{Maximize } \sum_k u_k y_{k,j0}$$
$$\text{Subject to } \sum_i v_i x_{i,j0} = 1$$
$$\sum_k u_k y_{k,j} \leq \sum_i v_i x_{i,j} \ \forall j \tag{3}$$
$$u_k, v_i \geq 0$$

Note that the decision variables are $u$ and $v$ as weights. In some cases, the dual model is preferable for some primal models that have many rows and columns. The dual DEA model can be represented as:

$$\text{Minimize } z_0 = \theta_{j0}$$
$$\sum_j \lambda_j y_{k,j} \geq y_{k,j0}$$
$$\theta_{j0} x_{i,j0} \geq \sum_j \lambda_j x_{i,j} \tag{4}$$
$$\lambda_j \geq 0$$

Model (4) will be used to compare the inefficient DMUs and efficient DMUs as the benchmark units and cross-efficiency will be determined the potential DMUs to become efficient after some adjustments in their inputs or outputs values.

The DEA model has been proposed in other forms. We discussed the CCR model above [50] as one of the standard DEA models.

### 3.2. Tobit Regression Analysis

Tobin [55] established the Tobit regression model, which is a statistical model based on linear assumptions that are employed when information on the dependent variable is unavailable for all observations due to censoring. The skewness of the continuous dependent variable to one side is the reason for altering part of the information. As a result, by altering, the regression is enabled. The population's conventional Tobit regression model is specified as:

$$
\begin{aligned}
y^* &= x + u \\
u &\sim N(0, \sigma^2) \\
y &= max(0, y^*)
\end{aligned}
\tag{5}
$$

$y^*$ is a vector of the dependent variable;
$x$ is a vector of the independent variable;
$\beta$ is a vector coefficient estimated by Tobit regression analysis;
$u$ is a vector of error in terms of normal distribution.

The Tobin regression model has been used to find the relationship between dependent and independent variables.

As part of this study, we analysed how some of the green principles' factors can affect the whole supply chain efficiency. The two factors that we are focusing on are environmental and economic. The three selected sub-factors are:

(1) Service Level: it's the percentage of the orders from customers that need to be satisfied and the time they need to wait to receive the service or product.
(2) Emission ($CO_2$): the average of each truck emission produced per day in kilograms.
(3) Arcs: the total number of arcs in each chain (size of the chain for transportation factor).

In this test, we considered the efficiency of each supply chain as a dependent variable and the three mentioned sub-factors like service level, emission, and arcs were considered independent variables to see their effects on the supply chain efficiency. Therefore, the following is used for Tobit regression analysis.

$$
efficiency = \beta_1(Service\ Level) + \beta_2(Emission) + \beta_3(Arcs) + u
\tag{6}
$$

The Tobit test will be provided with a better understanding of the changes in efficiency based on some factors of green principles.

With the goal of presenting the novelty of this research and better describing the process of introducing the approach, a sequence of steps is illustrated in Figure 1.

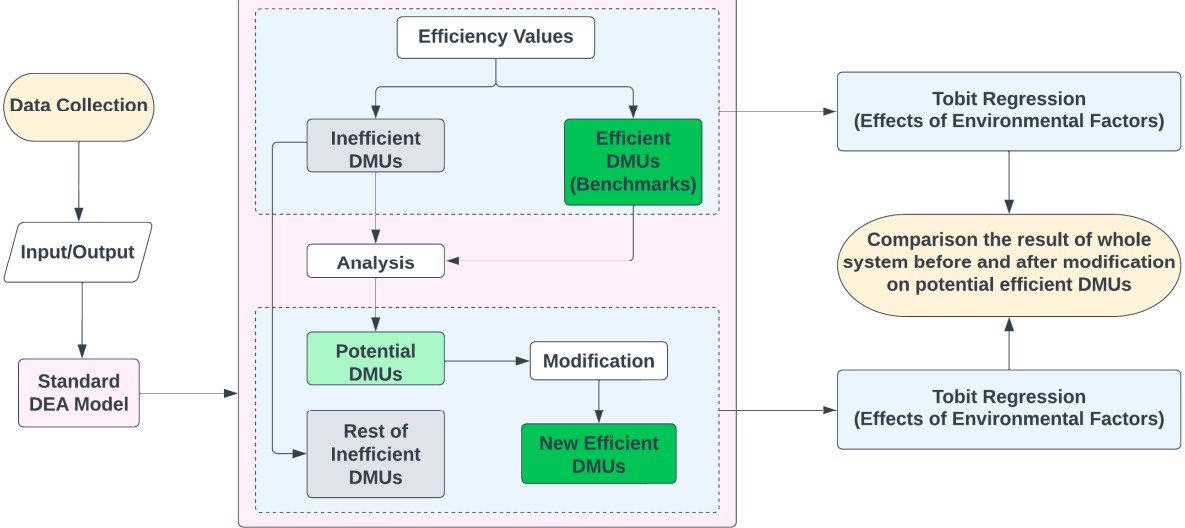

**Figure 1.** Methodology summary diagram.

## 4. Numerical Example

There are 38 supply chains from different companies that differ in size, material, and final product for applying the CCR model to find which of them can consider efficient DMUs (see Appendix A). Then, according to those who were selected as efficient, cross-efficiency upon the benchmarks was applied to find out which of the inefficient DMUs can become efficient. First of all, we categorized the data and specified the chains with their structure according to their final product profit. For this purpose, inputs including distribution, manufacture, supplier, retailer, and transportation and output as an average cost of goods sold (ACGS) for each chain were considered. Table 2 shows the minimum, maximum, average, and variance of the cost metrics for each stage and their percentage from the total ACGS.

**Table 2.** Summary of collected data for 38 under-evaluation DMUs.

|  | Dist. | Manuf. | Part. | Retail. | Transp. | Av. CGS |
|---|---|---|---|---|---|---|
| Min. | 0.0001% | 0.1% | 23.3% | 0.0001% | 0.1% | $3.12 |
| Max. | 64.2% | 100.0% | 99.1% | 6.0% | 62.8% | $150,816.00 |
| Av. | 8.9% | 29.5% | 70.6% | 1.8% | 6.7% | $8706.01 |
| Var. | 0.025675 | 0.082809 | 0.043919 | 0.0003146 | 0.021063 | - |

Adapted from [56].

### 4.1. DEA Model

The CCR input-oriented model as one of the standard DEA models is applied to the extracted data to find out which one of the DMUs is efficient. Table 3 shows the result of the CCR model and finding out the efficient DMUs. Model selection was made based on constant returns to scale assumption of under-evaluation of DMUs. This means that any radial increase in the input vector will have resulted in a proportional radial increase in the output vector. The model was running according to each ACGS chain because even the financial part is an essential part of each chain and management besides greening the supply chain.

**Table 3.** First, run the model result.

| DMU | Efficiency | DMU | Efficiency |
|---|---|---|---|
| 01 | 54.49 | 20 | 35.4 |
| 02 | 62.12 | 21 | 54.87 |
| 03 | 41.97 | 22 | 48.3 |
| 04 | 79.14 | 23 | 60.95 |
| 05 | 38.94 | 24 | 100 |
| 06 | 66.9 | 25 | 65.58 |
| 07 | 37.26 | 26 | 37.66 |
| 08 | 39.22 | 27 | 57.56 |
| 09 | 52.48 | 28 | 100 |
| 10 | 86.21 | 29 | 36.84 |
| 11 | 39.05 | 30 | 61.84 |
| 12 | 68.1 | 31 | 100 |
| 13 | 100 | 32 | 69.19 |
| 14 | 100 | 33 | 100 |
| 15 | 57.1 | 34 | 100 |
| 16 | 85.57 | 35 | 36.43 |
| 17 | 78.84 | 36 | 100 |
| 18 | 27.32 | 37 | 34.04 |
| 19 | 59.05 | 38 | 100 |

As obvious from the results of the CCR model in Table 3, there are nine efficient DMUs. The efficient DMUs are 13, 14, 24, 28, 31, 33, 34, 36, and 38. They are selected as benchmarks for cross-efficiency verification. Thus, the optimal weights of each efficient DMU are used to compute the relative efficiency value (Equation (1)) for each inefficient DMU.

Table 4 contains the cross-efficiency values for each inefficient DMU with respect to the benchmark DMUs. The eleventh column of this table shows that each one of the inefficient DMUs can be efficient, with respect to the optimal weights of the benchmark DMUs (the number of 100 in each row). A DMU can consider a potentially efficient DMU when its efficiency value is near 100 and/or its relevant efficiency value is 100 more than four times in cross-efficiency verification. Moreover, as a supportive factor for the above criteria, the summation of cross-efficiency values in each line compute for inefficient DMUs as the last column of Table 4, shows which one of the inefficient DMUs where more efficient according to the benchmark DMUs weights.

**Table 4.** Result of cross-efficiency verification.

| DMU | 13 | 14 | 24 | 28 | 31 | 33 | 34 | 36 | 38 | | Eff | Total |
|-----|------|-----|-----|------|------|------|------|------|-----|---|-------|--------|
| 01 | 100 | 2.23 | 18.67 | 100 | 34.57 | 1.27 | 100 | 100 | 20.35 | 3 | 54.49 | 477.09 |
| 02 | 100 | 100 | 22.65 | 27.97 | 43.58 | 9.7 | 18.1 | 74.4 | 100 | 3 | 62.12 | 496.4 |
| 03 | 100 | 8.21 | 100 | 55.3 | 7.17 | 100 | 70.66 | 29.75 | 100 | 3 | 41.97 | 571.09 |
| 04 | 100 | 13.95 | 100 | 86.45 | 56.48 | 15.95 | 67.97 | 100 | 100 | 4 | 79.14 | 640.8 |
| 05 | 74.24 | 3.31 | 100 | 100 | 100 | 3.26 | 100 | 100 | 28.17 | 5 | 38.94 | 608.98 |
| 06 | 100 | 10.73 | 64.52 | 44.83 | 9.07 | 100 | 41.72 | 34.93 | 100 | 3 | 66.9 | 505.8 |
| 07 | 100 | 7.39 | 100 | 61.35 | 6.53 | 33.95 | 100 | 27.87 | 100 | 4 | 37.26 | 537.09 |
| 08 | 100 | 8.21 | 100 | 55.3 | 7.17 | 100 | 70.66 | 29.75 | 100 | 4 | 39.22 | 571.09 |
| 09 | 78.25 | 14.27 | 100 | 89.8 | 100 | 14.79 | 70.8 | 100 | 100 | 4 | 52.48 | 667.91 |
| 10 | 100 | 11.06 | 100 | 100 | 24.2 | 12.35 | 100 | 70.68 | 100 | 5 | 86.21 | 618.29 |
| 11 | 100 | 100 | 22.65 | 27.97 | 43.58 | 9.7 | 18.1 | 74.4 | 100 | 3 | 39.05 | 496.4 |
| 12 | 100 | 8.5 | 100 | 55.54 | 7.58 | 100 | 67.97 | 31.05 | 100 | 4 | 68.1 | 570.64 |
| 15 | 78.25 | 14.27 | 100 | 89.8 | 100 | 14.79 | 70.8 | 100 | 100 | 4 | 57.1 | 667.91 |
| 16 | 100 | 13.95 | 100 | 86.45 | 56.48 | 15.95 | 67.97 | 100 | 100 | 4 | 85.57 | 640.8 |
| 17 | 100 | 13.95 | 100 | 86.45 | 56.48 | 15.95 | 67.97 | 100 | 100 | 4 | 78.84 | 640.8 |
| 18 | 100 | 13.95 | 100 | 86.45 | 56.48 | 15.95 | 67.97 | 100 | 100 | 4 | 27.32 | 640.8 |
| 19 | 100 | 7.39 | 100 | 61.35 | 6.53 | 33.95 | 100 | 27.87 | 100 | 4 | 59.05 | 537.09 |
| 20 | 100 | 7.39 | 100 | 61.35 | 6.53 | 33.95 | 100 | 27.87 | 100 | 4 | 35.4 | 537.09 |
| 21 | 100 | 3.48 | 100 | 100 | 50.07 | 3.48 | 100 | 100 | 29.91 | 3 | 54.87 | 586.94 |
| 22 | 100 | 100 | 31.75 | 30.23 | 61.41 | 41.34 | 19.65 | 81.67 | 100 | 3 | 48.3 | 566.05 |
| 23 | 100 | 2.23 | 18.67 | 100 | 34.57 | 1.27 | 100 | 100 | 20.35 | 4 | 60.95 | 477.09 |
| 25 | 78.25 | 14.27 | 100 | 89.8 | 100 | 14.79 | 70.8 | 100 | 100 | 3 | 65.58 | 667.91 |
| 26 | 100 | 2.23 | 18.67 | 100 | 34.57 | 1.27 | 100 | 100 | 20.35 | 4 | 37.66 | 477.09 |
| 27 | 74.24 | 3.31 | 100 | 100 | 100 | 3.26 | 100 | 100 | 28.17 | 5 | 57.56 | 608.98 |
| 29 | 100 | 10.4 | 20.42 | 100 | 37.86 | 2.48 | 75.34 | 100 | 100 | 4 | 36.84 | 546.5 |
| 30 | 100 | 100 | 30.61 | 28.79 | 40.64 | 54.91 | 18.91 | 71.43 | 100 | 3 | 61.84 | 545.29 |
| 32 | 100 | 8.21 | 100 | 55.3 | 7.17 | 100 | 70.66 | 29.75 | 100 | 3 | 69.19 | 571.09 |
| 35 | 100 | 7.39 | 100 | 61.35 | 6.53 | 33.95 | 100 | 27.87 | 100 | 4 | 36.43 | 537.09 |
| 37 | 100 | 100 | 30.61 | 28.79 | 40.64 | 54.91 | 18.91 | 71.43 | 100 | 3 | 34.04 | 545.29 |
| Total | 23 | 4 | 18 | 8 | 4 | 5 | 11 | 14 | 24 | | | |

Due to mentioned criteria, six DMUs can be considered as potentially efficient DMUs. These six DMUs are 4, 5, 10, 16, 17, and 27.

The potentially efficient DMUs could be able to become efficient with small adjustments in the value of their inputs. The target results of the CCR model provide the necessary changes in the inputs of the potential DMUs to make them efficient. Table 5 contains the target results of input-oriented CCR model.

**Table 5.** Target results of CCR model for potential DMUs.

| DMU | Input 1 (%) | Input 2 (%) | Input 3 (%) | Input 4 (%) | Input 5 (%) |
|---|---|---|---|---|---|
| 4 | −89.4 | −20.86 | −20.86 | 0 | 0 |
| 5 | −61.06 | −610.6 | −61.06 | −61.06 | 0 |
| 10 | 0 | −13.79 | −13.79 | 0 | 0 |
| 16 | −97 | −14.43 | −14.43 | 0 | 0 |
| 17 | −94.16 | −21.16 | −21.16 | −21.16 | −21.16 |
| 27 | −42.44 | −42.44 | −42.44 | −42.44 | 0 |

If an inefficient potential DMU keeps its output at the same level and decreases its inputs as given in Table 5 it can perform efficiently. For example, DMU 10 with a 13.79% decrease in the value of inputs 2 and 3 (manufacturer and supplier) can be able to become efficient.

### 4.2. Tobit Regression Model

In this part, after recognizing the potential DMUs and finding out each of them should improve in which inputs and how much to become efficient. Then, Tobit regression was used to verify the effects of the green factors that we considered before as service level, emission ($CO_2$), and arcs which express the size of the chain, on the efficiency before and after the modification of potential DMUs to see which one and how much can affect the total efficiency of the whole supply chain. The result of Tobit regression before recognising the potential DMUs is shown in Table 6 below:

**Table 6.** Result of Tobit regression (for all DMUs before modification of potential DMUs).

| Efficiency | Coefficient | Std. err. | $t$ | $p > \lvert t \rvert$ | [95% conf. | interval] |
|---|---|---|---|---|---|---|
| Service level | 3.058122 | 2.432441 | 1.26 | 0.217 | −1.879996 | 7.99624 |
| Emission ($CO_2$) | −0.1192653 | 0.2147632 | −0.56 | 0.582 | −0.5552578 | 0.3167272 |
| Arcs | 0.4362361 | 0.2238707 | 1.95 | 0.059 | −0.0182455 | 0.8907178 |
| _cons | −2.297785 | 2.335146 | −0.98 | 0.332 | −7.038383 | 2.442814 |
| var (e. Efficiency) | 0.0477093 | 0.0109453 | | | 0.0299458 | 0.07601 |

As we can see here from the efficiency column, the service level has the most positive effect to improve the efficiency of the DMUs and emission has the most negative effect on the DMUs' efficiency. On the other hand, the number of arcs in the supply chain network has a slightly positive effect to make a DMU more efficient. The numbers in the coefficient column show the changes in the efficiency value of DMUs with respect to a one-unit change on each green factor. For example, if $CO_2$ emission increases by a unit, the total efficiency value will decrease by 0.1193% or if the service level increase by one unit the total efficiency value by approximately increases 3%. Moreover, the effect of one-unit increases in under-evaluation green factors on total efficiency value is 0.0477. Obviously, there is an inverse reaction to green factors when the total efficiency value of the supply chain system is increased. Therefore, after necessary adjustments on input values of potential DMUs and by nature increasing the total efficiency value, there will be some changes in the effect of the green factors on the efficiency of the whole system. To see this, the first Tobit test ran

for a selected six potential DMUs to see the changes and effects of green factors on their efficiency. The result is shown in Table 7 below:

**Table 7.** Result of Tobit regression (for six potential DMUs).

| Efficiency | Coefficient | Std. err. | $t$ | $p > |t|$ | [95% conf. | interval] |
|---|---|---|---|---|---|---|
| Service level | 9.254766 | 6.93133 | 1.34 | 0.274 | −12.80382 | 31.31335 |
| Emission ($CO_2$) | 2.07091 | 0.7734005 | 2.68 | 0.075 | −0.3903959 | 4.532215 |
| Arcs | 3.377883 | 3.981558 | 0.85 | 0.459 | −9.29321 | 16.04898 |
| _cons | −8.676349 | 6.86242 | −1.26 | 0.295 | −30.51563 | 13.16294 |
| var (e. Efficiency) | 0.0125101 | 0.0072227 | | | 0.001992 | 0.0785645 |

This table illustrates the significant positive effect of service level, $CO_2$ emission, and arcs on the selected DMUs' efficiency values. It shows that efficiency improvement on these DMUs will be caused an improvement in their network and relations between the stages and customer satisfaction of their service level dramatically. However, on the other side, this will have a positive undesirable effect on $CO_2$ emission.

The next Tobit test is conducted for all of the DMUs after adjustment on inputs of the potential DMUs and improving their efficiency. The results are summarized in Table 8. For the new set of supply chains still like in Table 6, service level and $CO_2$ have more positive and negative effects on the total efficiency level of the system, respectively. On the other hand, the arcs have a slightly less positive effect. Moreover, the variety of the total efficiency value of the DMUs regarding the green factors is an increase from 0.047 to 0.058. This means that the efficiency scores of DMUs are now more sensitive to green factors.

**Table 8.** Result of Tobit regression (for all DMUs after modification of potential DMUs).

| Efficiency | Coefficient | Std. err. | $t$ | $p > |t|$ | [95% conf. | interval] |
|---|---|---|---|---|---|---|
| Service level | 5.13511 | 2.683682 | 1.91 | 0.064 | −0.3130544 | 10.58327 |
| Emission ($CO_2$) | −0.1338235 | 0.2369456 | −0.56 | 0.576 | −0.6148486 | 0.3472016 |
| Arcs | 0.2947164 | 0.2469937 | 1.19 | 0.241 | −0.2067075 | 0.7961404 |
| _cons | −4.233967 | 2.576338 | −1.64 | 0.109 | −9.464211 | 0.996276 |
| var (e. Efficiency) | 0.0580738 | 0.0133231 | | | 0.0364513 | 0.0925227 |

To verify which one of the potential DMUs has the best situation to invest in for improvement and become efficient besides ranking them based on their efficiency values, a priority verification is conducted again by Tobit test. The efficiency verity of each one of the potential DMUs are measured regarding the benchmark DMUs separately and the results are shows in Table 9.

**Table 9.** Result of Tobit regression for each potential DMU with efficient DMUs.

| Rank | DMU | var (e. Efficiency) | Std. err. | Efficiency |
|---|---|---|---|---|
| 1 | 10 | 0.0008452 | 0.000378 | 86.21 |
| 2 | 16 | 0.0011205 | 0.0005011 | 85.57 |
| 3 | 4 | 0.0016289 | 0.0007284 | 79.14 |
| 4 | 17 | 0.003404 | 0.0015223 | 78.84 |
| 5 | 27 | 0.0145833 | 0.0065218 | 57.56 |
| 6 | 5 | 0.0277673 | 0.0124179 | 38.94 |



The third column of the above table shows the effect of the green factors on the efficiency value of each one of the potential DMUs. Based on these numbers and their standard deviation of them DMU 10 has the best situation for efficiency improvement. As seen in Table 9, the mentioned ranking method is matching the priority given by the efficiency values of the DMUs.

Thus, DMU 10 is the best candidate among the six potential DMUs for investing in improving and adjusting to becoming efficient. In this case, because of the fewer effects of green factors principles, it can be efficient and more robust compared to the other potential DMUs.

## 5. Conclusions

This study attempts to write a general DEA model to distinguish the efficient (supply chains) DMUs from the under-evaluated DMUs, according to green supply chain management principles. In the first stage, the CCR model has been applied to 38 DMUs, and the result has shown that 9 DMUs are efficient. Then, the initial efficient DMUs considered as a benchmark for the second stage by applying cross-efficiency upon the elected ones to find out which DMUs from non-selected efficient have the potential to become efficient. According to the result, six DMUs can be efficient with a slight improvement in one or some inputs according to the target results of the CCR model (Table 5). After that, the Tobit regression model has been applied to investigate the effects of some green principal factors like service level, emissions ($CO_2$), and arcs to measure their effects on supply chain efficiency before modification of potential DMUs (Table 6). In general, the service level has the most positive effect on efficiency. By contrast, the emission has the most negative impact on the efficiency of the supply chains. Additionally, the number of arcs for each chain which is the network size has a slight positive effect on the supply chain efficiency. However, when the Tobit regression model is applied to the six potential DMUs, all three green factors had a positive effect on their efficiency improvement, even emission as an undesirable effect on the efficiency. After the modification of six potential DMUs as efficient, now there are 15 efficient DMUs in whole supply chains between 38. In the end, the Tobit regression model is used to find out the effect of total efficient DMUs on the whole supply chain's efficiency after modification of potential DMUs. Table 8 illustrates that the general idea about the green factors' effect on the total efficiency of the supply chains was correct but we should be aware of improving the efficiency of supply chains to hold the green principles factor efficiency robust.

Additionally, the six potential DMUs were compared with each other to recognize which of them has more value and a better situation to invest for improving its efficiency on it. It is found that DMU 10, according to the efficiency and standard deviation number, is the best candidate among the potential DMUs to invest for its efficiency improvement (Table 9).

As it was expected, service level (customer satisfaction) and arcs (as the size of the transportation factor) had a positive effect and $CO_2$ emission (environmental effect) had a negative effect on the efficiency value of whole DMUs. However, for potentially efficient DMUs, the effect of mentioned green factors was positive. This means that there is a benefit in the green aspect to improve the potential DMUs and make them efficient. Moreover, after necessary adjustment on potential DMUs, the Tobit regression results show that the sensitivity of the whole system is increased regarding the under-consideration of green factors. This indicates on importance of priority to modify the potential DMUs according to their ranks which are provided by the efficiency values and Tobit test results.

For future studies, we can use stochastic data in time for each stage can be used in the supply chain to find the efficient supply chain in the first phase, and in the second phase, more green principles can be considered to see their effect on the whole supply chain efficiency and see the difference of applying the green factors on calculation.

**Author Contributions:** Conceptualization, F.Z.T.; methodology, S.D.; software, F.Z.T. and S.D.; data curation, F.Z.T.; writing—original draft, F.Z.T.; writing—review and editing, S.D.; supervision, S.D. All authors have read and agreed to the published version of the manuscript.

**Funding:** This research received no external funding.

**Institutional Review Board Statement:** Not applicable.

**Informed Consent Statement:** Not applicable.

**Data Availability Statement:** The data that support the findings of this study are openly available in "Real-World Multi-Echelon Supply Chains Used for Inventory Optimization" by Sean P. Willems [56]. The data are open to all researchers as long as the researcher is willing to cite the published M&SOM paper as the source.

**Conflicts of Interest:** The authors declare no conflict of interest.

**Appendix A**

Figure A1 below illustrate the industries' chains with their size and the average cost of goods sold per unit of production. The characteristics of the 38-supply chain that we encompass in this study with their number of arcs (size of the chain) and ACGS are shown in Table A1.

**Table A1.** General classification of the dataset.

| Chain Name | SIC Description | Total Arcs (Size of the Chain) | Average Cost of Goods Sold Per Unit |
|:---:|:---:|:---:|:---:|
| 1 | Industrial Organic Chemicals, Not Elsewhere Classified | 10 | $71.88 |
| 2 | Semiconductors and Related Devices | 13 | $136.07 |
| 3 | Computer Peripheral Equipment, Not Elsewhere Classified | 18 | $3820.00 |
| 4 | Games, Toys, and Children's Vehicles, Except Dolls and Bicycles | 39 | $212.67 |
| 5 | Food Preparations, Not Elsewhere Classified | 31 | $31.46 |
| 6 | Cutlery | 28 | $3.42 |
| 7 | Construction Machinery and Equipment | 78 | $150,816.00 |
| 8 | Electromedical and Electrotherapeutic Apparatus | 48 | $120.72 |
| 9 | Cereal Breakfast Foods | 52 | $22.84 |
| 10 | Electrical Appliances, Televisions, and Radio Sets | 176 | $5477.00 |
| 11 | Construction Machinery and Equipment | 108 | $142,853.00 |
| 12 | Cereal Breakfast Foods | 107 | $29.61 |
| 13 | Semiconductors and Related Devices | 452 | $134.50 |
| 14 | Arrangement of Transportation of Freight and Cargo | 119 | $18.11 |
| 15 | Soap and Other Detergents, Except Specialty Cleaners | 164 | $9.17 |
| 16 | Electromedical and Electrotherapeutic Apparatus | 224 | $342.88 |
| 17 | Computer Peripheral Equipment, Not Elsewhere Classified | 211 | $33.16 |
| 18 | Computer Peripheral Equipment, Not Elsewhere Classified | 224 | $91.68 |
| 19 | Computer Peripheral Equipment, Not Elsewhere Classified | 263 | $135.40 |
| 20 | Computer Peripheral Equipment, Not Elsewhere Classified | 169 | $448.02 |
| 21 | Perfumes, Cosmetics, and Other Toilet Preparations | 359 | $17.81 |
| 22 | Pharmaceutical Preparations | 253 | $3.23 |
| 23 | Paints, Varnishes, Lacquers, Enamels, and Allied Products | 524 | $110.54 |
| 24 | Power-Driven Handtools | 1245 | $949.26 |
| 25 | Farm Machinery and Equipment | 853 | $2319.00 |
| 26 | Aircraft Engines and Engine Parts | 605 | $11,681.14 |
| 27 | Electromedical and Electrotherapeutic Apparatus | 941 | $72.80 |

**Table A1.** *Cont.*

| Chain Name | SIC Description | Total Arcs (Size of the Chain) | Average Cost of Goods Sold Per Unit |
|:---:|:---:|:---:|:---:|
| 28 | Computer Storage Devices | 2262 | $149.71 |
| 29 | Primary Batteries, Dry and Wet | 753 | $8.17 |
| 30 | Arrangement of Transportation of Freight and Cargo | 632 | $6.73 |
| 31 | Farm Machinery and Equipment | 908 | $9609.00 |
| 32 | Perfumes, Cosmetics, and Other Toilet Preparations | 1685 | $3.12 |
| 33 | Perfumes, Cosmetics, and Other Toilet Preparations | 1009 | $6.29 |
| 34 | Telephone and Telegraph Apparatus | 4063 | $130.79 |
| 35 | Electromedical and Electrotherapeutic Apparatus | 1857 | $6.54 |
| 36 | Farm Machinery and Equipment | 4812 | $422.44 |
| 37 | Industrial Organic Chemicals, Not Elsewhere Classified | 2069 | $231.75 |
| 38 | Aircraft Engines and Engine Parts | 16,225 | $292.52 |

Adapted from [56].

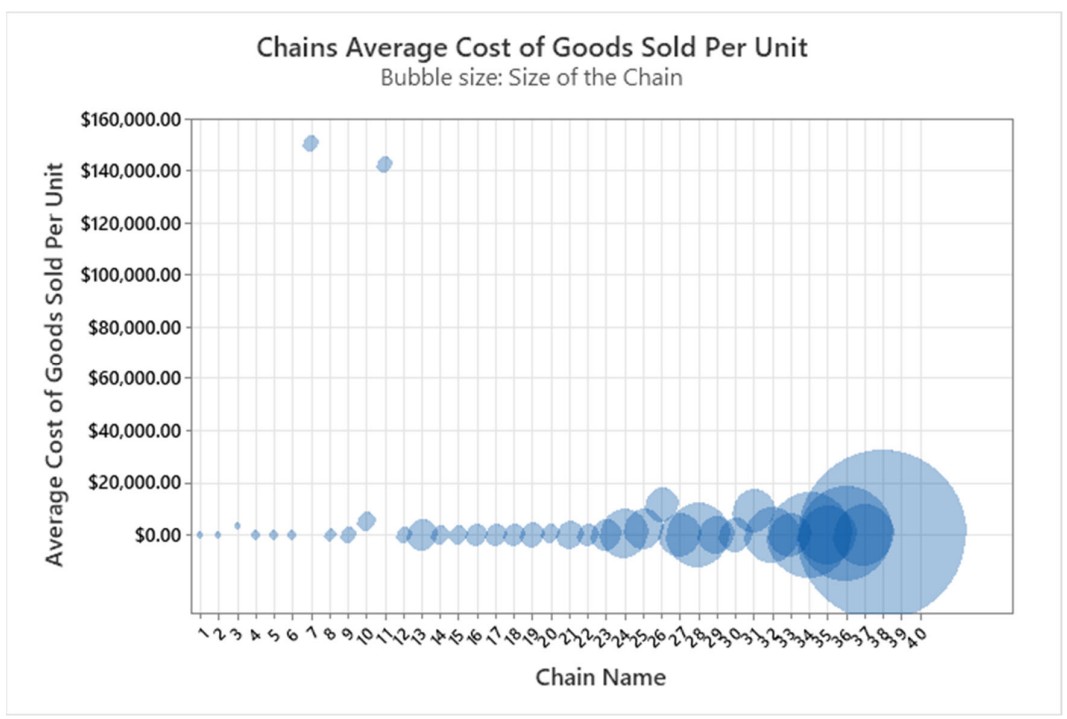

**Figure A1.** Summarizes each industry chain ACGS and Size.

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
