# Peer review of "Benchmark Approach for Efficiency Improvement in Green Supply Chain Management with DEA Models"

_sustainability, doi:10.3390/su15054433_

Round 1
Reviewer 1 Report
Thank you very much for reviewing the manuscript (sustainability-2198865). This research is designed to suggest a DEA-based model to evaluate the supply chain management efficiency. It can be improved by the following tips.
(1) The Abstract should be condensed by a minor background information.
(2) Literature review should be reworked by a clear logic. It is difficult to follow. Further, Table 1 should be revised by a concise description.
(3) Section 3 is about the models you adopted in this research. However, the relations between the two models are weak and should be revised to describe why they are combined for your research.
(4) The Section 4 is a case study. However, the results are difficult to reveal novel findings from this case study. The authors should have a careful and deep thinking from the results.
(5) Some editorial errors should be rejected. For example, the errors in Line 191 and 375.
Author Response
Dear Sir
We appreciate you for your precious time in reviewing our paper and providing valuable comments. Your valuable and insightful comments led to possible improvements in the current version. The authors have carefully considered the comments and tried our best to address every one of them. We hope the manuscript after careful revisions meet your high standards. The authors welcome further constructive comments if any.
We provide the point-by-point responses in the attached file. All modifications in the manuscript have been highlighted in red.

Reviewer 2 Report
Dear authors:
You have done a good job regarding the study's content, but some concerns arose regarding the format.
I believe you have two, not one article. I suggest splitting your article in two. The first should deal with DEA, and the second with the Tobit regression. As is, you have dispersed your attention in two focuses, and the result could be more precise. I suggest not withdrawing the article but submitting the new version dealing only with DEA. Later, you should write and submit the second article. Therefore, my issues embrace only the DEA study.
Your introduction needs to be better. You must contextualize the problem, derive a research gap from the extant literature, and answer a research question through the article. Your research question is: how can we measure efficiency in green supply chains? (or something similar).
The next issue regards your references. You employ too few references. Please enlarge much more, your reference basis. For instance, I suggest one and only one comprehensive reference regarding GSCM (please see https://www.mdpi.com/2071-1050/13/15/8127). You must check for additional references in databases. Please employ only peer-reviewed articles issued after 2016. As a rule of thumb, please consider one falsifiable statement, one required reference.
You should insert a table with the characteristics of the 38 SC you encompass in the study. Also, a demographic analysis (histograms) of the sample is due, focusing on four main variables (industry, size of the chain, annual revenue, and years of operation).
The discussion on the results could be better and should be enlarged to embrace the main implications of your study. It would be best if you explicitly answered who wins what and why upon your findings.
I believe you can easily amend your article and proceed to publication.
Best regards
Author Response
Dear Sir,
We appreciate you for your precious time in reviewing our paper and providing valuable comments. Your valuable and insightful comments led to possible improvements in the current version. The authors have carefully considered the comments and tried our best to address every one of them. We hope the manuscript after careful revisions meet your high standards. The authors welcome further constructive comments if any.
We provide the point-by-point responses in the attached file. All modifications in the manuscript have been highlighted in red.

Round 2
Reviewer 1 Report
The authors have concerned my reviews.
Reviewer 2 Report
Ok